# Effects of 8-Week In-Season Contrast Strength Training Program on Measures of Athletic Performance and Lower-Limb Asymmetry in Male Youth Volleyball Players

**DOI:** 10.3390/ijerph19116547

**Published:** 2022-05-27

**Authors:** Abdeltif Mesfar, Raouf Hammami, Walid Selmi, Sabri Gaied-Chortane, Michael Duncan, Thomas G. Bowman, Hadi Nobari, Roland van den Tillaar

**Affiliations:** 1Higher Institute of Sport and Physical Education of Ksar Said, Manouba University, Tunis 2010, Tunisia; mesfarabdellatif.ksarsaid@gmail.com (A.M.); raouf.cnmss@gmail.com (R.H.); selmiwalid13@yahoo.fr (W.S.); sabrigaied1@gmail.com (S.G.-C.); 2Research Laboratory: Education, Motor Skills, Sports and Health (EM2S, UR15JS01), Higher Institute of Sport and Physical Education of Sfax, University of Sfax, Sfax 3029, Tunisia; 3Research Unit (UR17JS01) “Sport Performance, Health & Society”, Higher Institute of Sport and Physical Education of Ksar Said, Manouba University, Tunis 2010, Tunisia; 4Centre for Sport, Exercise and Life Sciences, Coventry University, Coventry CV1 5FB, UK; aa8396@coventry.ac.uk; 5Department of Athletic Training, College of Health Sciences, University of Lynchburg, Lynchburg, VA 24501, USA; 6Faculty of Sport Sciences, University of Extremadura, 10003 Cáceres, Spain; 7Department of Exercise Physiology, Faculty of Educational Sciences and Psychology, University of Mohaghegh Ardabili, Ardabil 56199-11367, Iran; 8Department of Motor Performance, Faculty of Physical Education and Mountain Sports, Transilvania University of Braşov, 500068 Braşov, Romania; 9Department of Sports Science, Nord University, 7600 Levanger, Norway

**Keywords:** resistance training, power exercise, team sport, conditioning capabilities, lower extremity, dynamic balance

## Abstract

Strength training using high and lower load such as contrast training (CST) seems to be beneficial as it addresses larger adaptive reserves in youth athletes. Therefore, the aim of this study was to investigate the effects of CST on dynamic balance (composite score during dynamic balance test (CS-YBT)), one repetition maximum lower-limb back squat (1RM), jumping performance (single-leg hop (SLH) or countermovement jump height (CMJ)), lower-limb asymmetry (predicted from the single-leg jump performance between two legs [ILA]) in elite youth male volleyball players. Thirty-one male youth volleyball players aged 14 years were randomly assigned to a CST group (*n* = 16) or a control group (*n* = 15). The tests were performed before and after 8 weeks of training. Significant group × time interactions was observed for CS-YBT [*p* < 0.001, η_p_^2^ = 0.70], 1RM [*p* < 0.001, η_p_^2^ = 0.95], SLH with right and left leg [*p* < 0.001, η_p_^2^ = 0.69 and 0.51], CMJ [*p* < 0.001, η_p_^2^ = 0.47]), whilst it was not notable in ILA [*p* < 0.294]. Post hoc tests showed that CST group demonstrated greater improvement in all of the dependent variables from medium to large effect size (for all *p* < 0.001). As a result, 8 weeks of CST twice a week can be an effective and efficient training along with volleyball training to improve skill-related fitness measures, except for lower-limb asymmetry in young volleyball players.

## 1. Introduction

The performance of volleyball players is related to their ability to produce explosive actions in attack and block situations, as the goal of the game is to make the ball cross the 2.43 m net (for males) and hit the ground, while maintaining control of balance [1,2]. For instance, in highly dynamic situations in volleyball, rapid vertical jumping, proper dynamic alignment of the center of pressure relative to the base of support is essential for successful performance [3,4]. Furthermore, due to the importance of balance and muscular strength, for necessary power development while jumping during sport participation and volleyball training [5]. In addition, volleyball players’ jump performance might be hampered by strength imbalances in the lower limbs [1]. Therefore, identifying optimal resistance training methods to increase balance, muscle strength and power performance, and reduce asymmetry may be crucial to volleyball training and, indeed, performance.

Contrast training is commonly utilized by volleyball players during resistance training. Contrast strength training (CST) is characterized by the use of high and low loads in the same training session [6,7] and has previously been shown as an effective modality in improving both muscle strength and power, with adaptations in both neuromuscular function and muscle morphology [8]. For example, Smilios et al. [7] showed that contrast training program led to increases in vertical jump, sprint and agility levels in prepubertal children who exhibit high muscular strength trainability. Furthermore, Hammami et al. [6] demonstrated that both contrast strength and plyometric training programs enhanced muscle power and sprint performance at 5 and 40 m distances and change-of-direction test scores relative to controls in male youth soccer players. The authors concluded that the improvement of physical performance was greater following eight weeks of contrast strength training than with plyometric training. Whilst current research has endeavored to verify the effectiveness of contrast training on enhancing athletic performance with youth, there is a shortage of research exploring the dose–response relationship of such modality in volleyball.

In addition to performance, lower-limb asymmetries (LLA) have frequently been studied to quantify performance differences between limbs [9,10,11,12] and a bilateral asymmetry in youth [1]. Madruga-Parera et al. [13] demonstrated that contrast resistance training programs reduced interlimb asymmetries score (−0.70 moderate vs. −0.32, small), more than isoinertial resistance training in youth male handball players. While some level of asymmetry is to be expected in youth populations using both isoinertial and contrast resistance training programs, further examination of the effects contrast training between-limb differences using practically viable screening tasks is warranted. Our starting point was the fact that contrast training provides broader neuromuscular adaptations, which result in greater transfer to a wide variety of performance variables [1,6,7,13]. Cumulatively, there is a dearth of available evidence to compare the effects of CST, followed by a detraining period, on measures of dynamic balance control and lower-limb asymmetry during field-based tests with youth volleyball players. Therefore, the aim of this study was to examine the effect of eight-week CST on measures of dynamic balance performance, muscle strength and power performance and lower-limb asymmetry in youth male volleyball players.

Based on earlier longitudinal studies, we hypothesized that the CST program would result in athletic performance improvements [6,7,8] and reduce the LLA scores [13], compared to a control group, in young volleyball players.

## 2. Materials and Methods

### 2.1. Study Design

To assess the effects of an 8-week, in-season CST program on dynamic balance, muscle strength and power performance, and lower-limb asymmetry in male youth volleyball players, a randomized pre- and post-test group design with a CST and a regular volleyball training (control) group was used. In the present study, participants were randomly allocated to one experimental group and a control. Group allocation was realized by adjusting for, age, maturation and their performance in the CMJ and 1RM of the study sample. In addition, the order of each trial was changed randomly between participants, in order to avoid learning effects and fatigue. The dependent variables were dynamic balance, muscle strength (1RM) and power (CMJ, SLH), and LLA.

### 2.2. Participants

Thirty-one male youth volleyball players, belonging to a first division Tunisian volleyball club (Club Sfaxien, Sfax, Tunisia), were recruited in this study. All players were randomly assigned to a CST program (*n* = 16; age: 14.4 ± 0.6 years; height: 181.8 ± 6.6 cm; body mass: 68.5 ± 11.1 kg; maturity offset: +1.49 ± 0.63 years) or a control group (*n* = 15; age: 14.5 ± 0.5 years; height: 180.1 ± 2.9 cm; body mass: 67 ± 5.7 kg; maturity offset: +1.36 ± 0.43 years). All participants had the same daily school and volleyball team-training schedules. They all had been playing volleyball on a regular basis three–four times a week (i.e., ∼90 min per session), with a match played during the weekend, for more than 3 years. Maturity offset of participants was calculated according to Moore et al. [14].

Before experimental testing, the study was conducted according to the Declaration of Helsinki, and the protocol was fully approved by the Ethics Committee of the National Centre of Medicine and Science of Sports of Tunis (CNMSS-LR09SEP01) before the commencement of the assessments. Written informed consent was obtained from parents/legal representatives of all participants before the commencement of the study.

### 2.3. Procedure

All procedures were performed during the second half of the competitive volleyball season (February and March 2020). Before any data were collected, all athletes participated in two orientation sessions to familiarize themselves with the experimental procedures to minimize any learning effect of testing. Participants were assessed for balance (CS-YBT), muscle strength (1RM half squat) and power (single-leg hop and counter movement jump) and lower-limb asymmetry, respectively, before and after eight-week CST program. Procedures were undertaken after a general warm-up that consisted of running, calisthenics, and stretching.

#### 2.3.1. Dynamic Balance

Dynamic balance performance was evaluated by the Y-Balance Test [14]. All trials were conducted barefoot. Participants stood on the dominant leg, with the most distal aspect of their big toe on the center of the footplate from the YBT Kit. The participants were then asked to push the reach-indicator block with the free limb in the anterior, posterior medial, and posterior lateral directions in relation to the stance foot on the central footplate, while maintaining their single-limb stance. [14]. Data collection followed the protocol of Kang et al. [15], where participants were not allowed to lift their heel during the Y-Balance test. The allowance of heel lift could reduce the importance of ankle dorsiflexion, while requiring the heel to maintain contact with the ground would emphasize ankle range of motion [16]. Maximal reach distances were recorded to the nearest 0.5 cm marker on the Y-balance test kit (Figure 1). The trial was repeated if the participant failed to maintain a unipedal stance, or failed to return the reaching foot to the starting position. A composite score [CS-YBT (%)] was calculated using the formula by summing the three distances covered divided by three times the length of the leg and multiplied by 100. Leg length was measured from the anterior superior iliac spine to the most distal part of the medial malleolus by using a tape measure while the subject laid in supine position [14,17]. In the present study, an excellent reliability score was reported for the CS-Y balance test with the intra class correlation coefficient (ICC) value of 0.93.

#### 2.3.2. Dynamic Strength

Muscle strength was assessed with a 1RM squat as reported by Keiner et al. [18]. Before attempting the 1RM, participants performed three sub-maximal sets of 1–6 repetitions with a light-to-moderate load (40 to 50% 1RM). Participants then performed a series of repetitions with an increased load. The increments in resistance were dependent on the effort required for the lift and became progressively smaller as the players approached their 1RM. Failure was defined as a lift falling short of the full range of motion on at least two attempts, spaced at least two minutes apart. The 1RM was then determined within 5 to 6 trials. Test–retest reliability was excellent for 1RM with an ICC of 0.98 with the present study.

#### 2.3.3. Single-Leg Hop Test

Single-leg hop tests have been shown to evaluate lower-limb power performance requiring slow stretch-shortening-cycle action in accordance to the described protocol by Ramirez-Campillo et al. [19]. The test was executed using a 5 m fiberglass metric tape affixed to a wooden floor. Players were instructed to use their arms to aid in the jump phase, using a one-foot stand (right and left), and perform a fast movement (approximately 120° knee flexion angle) followed by a jump for maximal distance. All players were instructed to land in an upright position during the jumps and to flex their knees after landing. The test was executed three times for each leg, with the starting order of the right or the left leg randomly assigned, with a 1 min rest period, and the best value was recorded for analysis. In the present study, test–retest reliability scores have been shown to be good with an ICC of 0.81.

Bilateral asymmetry was calculated from the performance measure during the single-leg hop test. A negative sign (−) was arbitrarily assigned when the left leg was the stronger one, and a positive sign (+) was used when the right leg was the stronger one. In the literature [20], relative inter-limb asymmetry for the lower limbs was determined the formula: (Right leg − Left leg)/(Right leg + Left leg) × 100. With this formula, it seems that index number 10 is more suitable for calculating the LLA between the two legs during the single-leg hop test. Test–retest reliability scores for LLA measures from the present results have been shown good with our study (ICC = 0.73).

#### 2.3.4. Vertical Jump Test

Subjects were instructed to keep an upright standing position until an angle of 90° knee flexion and perform a vertical jump. Players were performed to perform the jump with a quick manner in order to maximize their performance. During the test, two trials were carried out, with two minutes of passive recovery. The highest jump height performance was used for analysis. Test–retest reliability has been reported good for the CMJ test with an ICC value of 0.89 with the present study.

#### 2.3.5. Training Program

After the pretest, participants were randomly assigned to a perform CST or to a control group. Groups were matched for anthropometrics and physical characteristics. The groups did not differ in measures of the pre-test. The CST group performed the eight-week in-season training program consisting of strength exercises, including bench press, pull over, half squat and forward lunge (Table 1). These exercises were included in the study based on the muscle groups solicited in volleyball game and training. Sessions were performed twice weekly on non-consecutive days (Tuesday and Thursday).

The CST program used free weight resistance training exercises at 40 to 80% of 1-repetition maximum with 3–4 sets of 2–4 repetitions. The training was based on performing a heavy load lift followed by a light load lift, in the same series and the same exercise. Participants of the control group followed their standard volleyball practice over the same duration with no strength training design. It is important to note that all participants from the two groups had regularly performed strength/power training exercises (i.e., bench press, pull over, squat, forward lunge snatch, and plyometric) during competitions and training for a minimum of 2 years before the start of the study. The volume of training remained constant for all exercises. During this period, the control group was exposed to a 15 min period of passing drills. Of note, training volume (i.e., total time of training exposure) was similar between the two groups. Qualified coaches and experienced sports scientists supervised both groups. Throughout all training exercises, the instructor-to-player ratio of 1:1 was maintained. All subjects received treatment conditions as allocated. A standardized 10 min warm-up containing jogging, dynamic stretching exercises, calisthenics, and preparatory exercises (e.g., fundamental weightlifting exercises specific to the training program) was provided for all experimental groups before the beginning of each training session. The training session lasted ~35 min and ended with 5 min of cool down activities including dynamic stretching. All groups performed regular volleyball practice during 5 to 6 sessions throughout the study and no injuries were obtained over the training program.

### 2.4. Statistical Analysis

Data are presented as means and standard deviations (SD) and normality was assessed and confirmed using the Shapiro–Wilk test. The data were then analyzed using a 2 (groups: CST and control group) by 2 (time: pre, post) analysis of variance (ANOVA) for repeated measures. Where the assumption of sphericity was violated, Greenhouse–Geisser correction was used to interpret the results. If group × time interactions reached the level of significance, post hoc tests, using Bonferroni corrections, were computed to identify the comparisons that were statistically significant. Partial eta-squared (η_p_^2^) was used as a measure of effect size. ES can be classified as small (η_p_^2^ = 0.01), medium (η_p_^2^ = 0.06), or large (η_p_^2^ = 0.14) [21]. Test–retest reliability was assessed using the ICC and the standard error of measurement (SEM) expressed as coefficient of variation [21]. For the interpretation of ICC values, a value greater than 0.80 reflects an excellent reliability, whereas ICCs from 0.70 to 0.79 reflect a good reliability [22]. The alpha level of significance was set at *p* < 0.05. All data analyses were performed using SPSS 26.0 (SPSS, Inc., 288 Chicago, IL, USA).

## 3. Results

All 31 young volleyball players from CST and control group completed the study according to the study design and methodology. Participants attended all training sessions, and none reported any training- or test-related injury. There were no statistically significant between-group baseline differences found for any of the analyzed parameters (Table 2).

### 3.1. Dynamic Balance

Statistical calculations revealed a significant group × time interaction (*p* < 0.01, η_p_^2^ = 0.703) for CS-YBT. Bonferroni post hoc comparisons indicated significant increase from pre- and post-test were significant for both groups (both *p* = 0.001). Percent changes in ages (Δ) in CS-YBT between pre- and post-test were significantly greater in CST group (Δ = 11.7) than control group (Δ = 2.0).

### 3.2. RM Test

In view of the 1RM test, a significant group by time interaction was observed (*p* = 0.02, η_p_^2^ = 0.945). Bonferroni post hoc pairwise comparisons indicated significant differences at pre- and post-test between groups (*p* = 0.001). The increase pre- and post-test was significant for both intervention (*p* = 0.001) and control groups (*p* = 0.003) but the magnitude of change was greater for the intervention group (Δ = 18.8) compared to the control group (Δ = 2.2).

### 3.3. Single-Leg Hop Test

A significant group × time interactions were also noted (*p* < 0.01, η_p_^2^ = 0.693) (Figure 2). Bonferroni post hoc pairwise comparisons indicated significant differences at pre and post between groups (*p* = 0.001). The increase pre and post was significant for both intervention and control groups (both *p* = 0.001), but the magnitude of change was greater for the intervention group (Δ = 20.6) compared to the control group (Δ = 2.8).

Similarly, a significant group × time interactions was found for the SLHL (*p* < 0.02, η_p_^2^ = 0.511). Bonferroni post hoc pairwise comparisons indicated significant differences at pre and post between groups (*p* = 0.001). The increase pre and post-test was significant for both intervention and control groups (both *p* = 0.001), but the magnitude of change was greater for the intervention group (Δ = 11.7) compared to the control group (Δ = 2.8)

### 3.4. Countermovement Jump Height

A group-by-time interaction was observed significant for the CMJ height (*p* < 0.02, η_p_^2^ = 0.473) (Figure 3). Analyses demonstrated a medium improvement in CMJ height before and after training in favor of the CST group (*p* < 0.01, η_p_^2^ = 0.158). Bonferroni post hoc pairwise comparisons indicated no significant differences at pre-test between groups (*p* = 0.88) but a significant difference post between groups (*p* = 0.01). The increase at pre- and post-test was significant for both intervention and control groups (both *p* = 0.01), but the magnitude of change was greater for the intervention group (Δ = 6.2) compared to the control group (Δ = 1.0).

### 3.5. Interlimb Asymmetry

There was no significant group × time interaction (*p* = 0.29) but there was a significant main effect for ILA (*p* = 0.01, η_p_^2^ = 0.331). Bonferroni post hoc pairwise comparisons indicated a significant increase in ILA from pre- to post-test (6.2 ± 1.7 vs. 10.2 ± 1.6, mean diff = 3.968).

## 4. Discussion

The aim of this study was to compare the effect of 8-week in-season CST on dynamic balance, muscle strength and power performance and lower-limb asymmetry scores in youth male volleyball players. The main findings were that the CST group improved in all physical fitness parameters, compared to the control group, with the exception of lower-limb asymmetry.

The results demonstrated an improvement of dynamic balance performance in the CST group. Resistance training is an effective and safe mode of training for children and adolescents with positive effects on balance and stability [2,23,24]. Although balance and coordination performance are not yet mature in pediatric populations [25], it is possible that greater enhancement of dynamic balance performance with contrast strength exercises could lead to a greater muscle power output in this population. Particularly in the CST group, the coordinated postural control to mobilize a load with an important range of motion and with a force production necessitates higher level of balance and muscle strength and power capabilities. In the current study, CST was performed under less stable conditions with high-speed dynamic contractions performed within a more limited base of support or with the center of gravity being moved outside the base of support, which would be affected to a much greater extent by balance and strength/power output. Thus, the participants in this study positively responded to these balance stressors with demonstrable enhancement in dynamic balance performance, to a greater extent with contrast training.

The present results demonstrate that 8 week of contrast training resulted in an improvement in 1RM half squat and muscle power performance (i.e., CMJ and SLH) in the CST group to a greater extent than the control group. The results of the present study are in agreement with prior investigations which have found an enhancement of 1RM half-squat and power performance after contrast strength training [1,6,26,27]. Vissing et al. [27] suggested that power training-induced improvements in all 3 tests of maximal strength (leg extension; knee extension; and hamstring curl), and demonstrated these improvements during shorter training periods or higher initial training status of participants. Because youth athletes rely less on their glycolytic metabolism [2], have a weak hypertrophic response [24] and lower type 2 fiber composition [2] in comparison with adults, it has been suggested that high-speed strength and power training programs, such as contrast training, may benefit youth athletes to enhance muscle strength and power performance. Accordingly, the effects of CST thus seem primarily due to neuromuscular adaptations, such as more effective motor unit recruitment, rate coding (frequency or rate of action potentials), synchronization, and intermuscular coordination [27,28].

Our results demonstrate that 8 week of CST did not reduce LLA scores in youth volleyball players. However, the range of LLA in the present study were low, and lower than the previously reported figures for runners of ≥10% [29,30]. The training process for volleyball in youth players emphasizes movement in a symmetrical manner. Coupled with the age of the youth participants involved, it is less likely that asymmetrical movement patterns had been developed, and therefore, we would not necessarily expect to observe a difference in LLA for this group because of the training stimulus that was applied. Given that balance and coordination are not fully developed in youth athletes [25], the implementation of 8 weeks of CST in youth resistance training, as implemented in the current study, should not be considered effective in reducing LLA in youth volleyball players.

Although the results of the present study should be considered a novel addition to the literature, this study is not without limitations, which should be mentioned accordingly. First, the results obtained should only be generalized to similar samples of participants. Furthermore, for a better understanding of underlying physiological mechanisms of adaptations associated with CST, a longer intervention duration, along with the evaluation of neuromuscular and muscle adaptations using electromyography, ultrasound, or other imaging technology, should be employed. Another limitation is that this study did not compare or contrast with plyometric and traditional strength training. Such a comparison would be useful in future research because previous investigations have indicated that differences in the adaptive responses between these methods may exist. Second, the sample size did not allow for participants to be grouped according to maturity status. This should be controlled for in future studies. Finally, the sample size of each groups was small. Therefore, this study is preliminary. However, it is difficult and almost impossible to recruit large sample sizes in elite sport, especially in a highly professionalized elite sport such as volleyball.

## 5. Conclusions

The current study shows that in youth volleyball players, after 8-week, the CST group had larger improvements in dynamic balance and muscle strength and power performances, to a greater extent than the control group. However, CST was not effective in enhancing LLA. Combining a specific stimulus, such as CST, into training sessions seems to be a safe training modality in this age cohort and facilitates continued progressive neuromuscular adaptation. Practitioners should include specific strength exercises, such as combining strength and power exercises with a progressive overload using CST to optimally enhance athletic performance in youth volleyball players. On that basis, coaches and key stakeholders of youth athletes are advised to add CST to their training routines with a view to maintain, and enhance, athletic performance in an optimal fashion to reduce LLA. This suggests that there could be interdependent positive transfer effects, from training that is singularly focused on strength and conditioning. Despite this, the present findings are based on longitudinal data which does conclusively allow for cause and effect to be determined. Furthermore, the findings suggest that when designing training programs aimed to improve athletic performance in volleyball players, coaches should pay attention to this specific adaptation, which can necessitate an individualized approach to program design.

### Practical Applications

The result of the CST intervention shows that youth volleyball players had larger improvements in dynamic balance and muscle strength, power performances, but not in LLA after 8 weeks.CST seems to be a safe training modality in youth age cohorts, and facilitates continued progressive neuromuscular adaptation.Coaches and key stakeholders of youth athletes may add CST to their training routines with a view to maintaining, and enhancing, athletic performance in an optimal fashion to reduce LLA.

## Figures and Tables

**Figure 1 ijerph-19-06547-f001:**
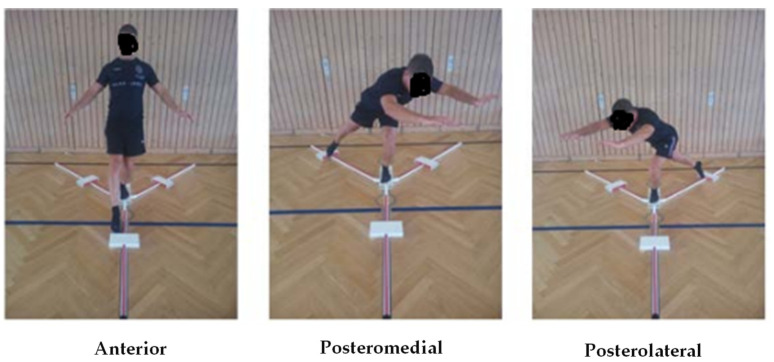
The Y-Balance test.

**Figure 2 ijerph-19-06547-f002:**
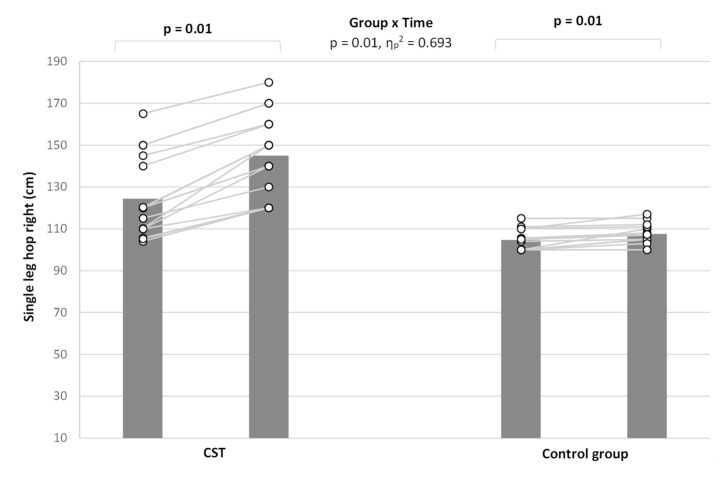
Participants performance before and after intervention period of single-leg hop right performance by the two groups. CST: contrast strength training group.

**Figure 3 ijerph-19-06547-f003:**
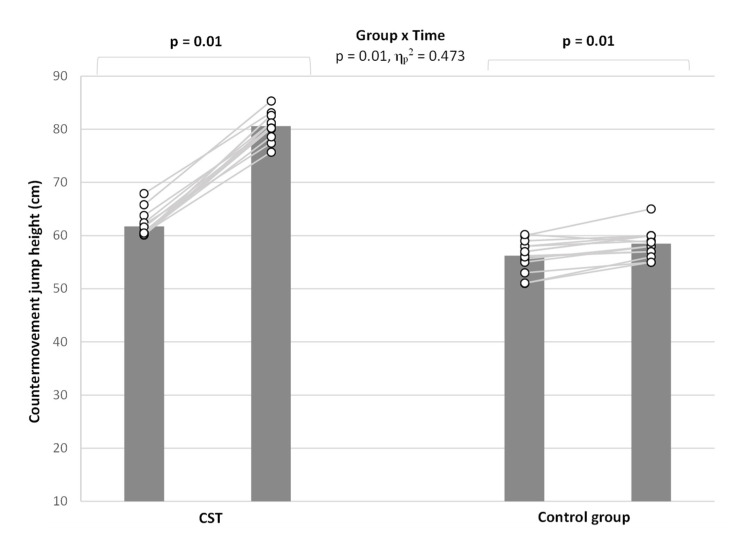
Participants performance before and after intervention of CMJ performance by the two groups.

**Table 1 ijerph-19-06547-t001:** Design of the 8-week contrast strength training program.

	CST
**Weeks**	**(% 1RM) for All Exercises**
**1–2**	3 sets × (2 reps at 70% 1RM + 4 reps at 40%)
**3–4**	4 sets × (2 reps at 80% 1RM + 4 reps at 50%)
**5–6**	3 sets × (2 reps at 70% 1RM + 4 reps at 40%)
**7–8**	4 sets × (2 reps at 80% 1RM + 4 reps at 50%)

Notes: CST = contrast strength training.

**Table 2 ijerph-19-06547-t002:** Effects of contrast strength training on measures of athletic performance in youth.

Variables	Groups	Pre-Intervention	Post-Intervention	Δ	ANOVA *p*-Value		
					Time	Group	Group × Time
**Composite score Y-balance test (%)**	**CST**	82.10 (5.19)	93.90 (3.36)	11.8	0.01	0.01	0.01
	**Control**	76.80 (5.51)	78.85 (5.37)	2.0			
**One repetition maximum (kg)**	**CST**	61.71 (2.28)	80.57 (2.26)	18.8	0.01	0.01	0.01
	**Control**	56.22 (2.92)	58.48 (2.59)	2.2			
**Single-leg hop test right leg (cm)**	**CST**	124.39 (19.23)	145.00 (19.66)	20.6	0.01	0.01	0.01
	**Control**	104.69 (5.09)	107.52 (5.09)	2.8			
**Single-leg hop test left leg (cm)**	**CST**	112.50 (4.65)	124.27 (4.11)	11.8	0.01	0.01	0.01
	**Control**	104.36 (5.85)	108.789 (5.91)	2.8			
**Countermovement jump height (cm)**	**CST**	32.69 (3.52)	38.94 (3.96)	6.2	0.01	0.03	0.01
	**Control**	32.50 (3.65)	33.50 (3.91)	1.0			
**Inter limb asymmetry (%)**	**CST**	7.83 (12.55)	12.94 (10.94)	5.1	0.01	0.19	0.29
	**Control**	4.69 (5.09)	7.52 (5.09)	2.8			

CST = contrast strength training.

## Data Availability

The data presented in this study are available on reasonable request from Abdeltif Mesfar. Requests for access to data should be sent to mesfarabdellatif.ksarsaid@gmail.com.

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
