# Peer review of "Effects of 8-Week In-Season Contrast Strength Training Program on Measures of Athletic Performance and Lower-Limb Asymmetry in Male Youth Volleyball Players"

_ijerph, 2022, doi:10.3390/ijerph19116547_

Round 1

Reviewer 1 Report

Thank you for the invitation to review for IJERPH. The submitted manuscript examines the effects of in-season contrast strength training in male youth volleyball players. The main outcomes comprise physical fitness variables (dynamic balance, strength, power) and lower limb asymmetry. The results demonstrate greater improvements for the intervention group, indicating its benefit when performed in addition to volleyball training. Overall, this is a well-written and interesting paper that might fit well within the scope of the Special Issue "Training for Optimal Sports Performance and Health". However, I have some major and several minor remarks regarding the manuscript and look forward to receiving a revised version by the authors.

General/ major comments:

  1. I recommend that data should be made available to the reviewers and to the readers, too. Please consider using an open access repository for data storage.
  2. It seems that no compulsory adjustments for multiple comparisons have been made in the statistical analyses (e.g., Bonferroni corrections to avoid alpha error accumulation). This must be addressed when revising the manuscript.
  3. It would be helpful if the authors could add a documentation of the procedures (particularly the Y-Balance test) as supplementary material.
  4. Using the reference provided below, I recommend a check whether the appropriate formula was used for the calculation of lower limb asymmetry.

Abstract:

  • Line 27 (and throughout the manuscript): Regarding the variables examined, I suggest being more specific: The YBT is a dynamic balance test - not just a (static) single-leg stance. Therefore, my recommendation is to rephrase “balance” to “dynamic balance”, also consistently throughout the manuscript.
  • Line 26-29 (and throughout the manuscript): Likewise, the authors categorize their variables by referring to the various motor abilities (dynamic balance, power…). Here, I recommend rephrasing the following sentence to become more specific:

“The tests were performed before and after 8-week training including dynamic balance (composite score during-balance test [CS-YBT]), maximum strength (one repetition maximum back squat, 1RM), explosive power (single leg hop [SLH] or countermovement jump height [CMJ]), as well as lower-limb asymmetry predicted from the single-leg jump performance between two legs [ILA]).”

  • Lines 30-31: Here, p-values with four decimal places are given; however, three decimals are sufficient. In the results section, only two decimals are presented. Please be consistent with three decimals for p-values throughout the manuscript.

Introduction

  • Line 66: Please elaborate on the term neuromuscular inter-limb asymmetries. What is meant here specifically, and what exactly was examined in the cited studies? In the current version, this does not become entirely clear to the readers. This is important, as these explanations lead to the objectives of the study.

Materials and Methods

  • Lines 96-97: Please explain, which kind of randomization has been carried out.
  • Line 171: In this respect, the authors indicate that groups were matched for anthropometric and physical characteristics. Please specify this procedure and describe it in more detail – thank you.
  • Line 100: How did you assess “socioeconomic backgrounds” – and is this likely to affect the results of the study? In case it is important to mention, please indicate a) how it was assessed, and b) the results of the assessment.
  • Lines 95-111: Please move all results (means & SD, table 1, ICC) to the results section.
  • Line 122: Is [13] the correct reference, here? Please check.
  • Lines 125-126: When a measuring tape is used, I doubt that the Y-balance test was used, as this is performed on the Y-balance board, on which participants shove a box in three directions and the read the distance reached on the scale of the Y-balance board. Hence, no measuring tape is needed. Maybe the Star Excursion Balance Test was used? Please clarify, ideally using a figure to describe testing procedures (maybe as supplementary material) – thank you!
  • Line 129: Please indicate how you measured the length of the leg?
  • Line 149: Please check the term “maximal height jump”. I think “jump for maximal distance” is more appropriate here.
  • Lines 155-161: For the assessment of bilateral asymmetry, the formula
    (A-B) / A x 100 is used and an appropriate reference is provided. However, I’d recommend consulting the paper by Parkinson et al. 2021: https://www.jssm.org/jssm-20-594.xml%3Eabst, particularly Table S1; to verify whether the formula used by the authors, corresponding to index nr. 7 in the above-mentioned paper, is more appropriate, or whether index nr. 10 seems more suitable for their study.
  • Lines 204-205: Please add the categories for ICC interpretation (e.g. low, moderate, high, excellent).
  • Line 168: In this respect, an ICC value of 0.89 indicates good reliability, and coefficients above 0.90 indicate excellent reliability? Please re-check.
  • Lines 202-203: What is the rationale for converting effect sizes from partial eta-squared to Cohen’s d? Partial eta-squared is a universally recognized/ accepted effect size, too. Please clarify.
  • Lines 195-207: It seems that no corrections for multiple comparisons (e.g. Bonferroni adjustments) have been applied. However, this seems to be necessary due to multiple testing. Please re-check and either apply these adjustments in the statistical analyses or – if already done – please indicate which procedure was used to avoid alpha error accumulation.

Results

  • Lines 248 & 258: Is it possible to improve the quality of the figures? Furthermore, effect sizes are presented as d and also as ES; and decimals of p-values vary from 2 to 3 places; but should be consistent.

Discussion

  • Lines 279-280: Please re-check “With children’s, balance and coordination are less developed [23].
  • Lines 284-286: I recommend rephrasing this sentence more cautiously. This might be an explanation for the findings of this study. However, “stress on the muscle spindles” was not subject to the investigation. At least, a reference supporting this statement/ explanation should be added.
  • Line 307: Again, I recommend the paper by Parkinson et al. 2021., who caution against the use of arbitrary asymmetry thresholds, such as 10-15%. I suggest adding a subordinate clause, referring to the failure to reference the origin of the evidence for an asymmetry threshold. Rather, more individual approaches to asymmetry have been proposed in recent studies recognizing the task-, metric- and population-specific nature of asymmetry.

References:

  • Lines 374-436: Please include the digital object identifier (DOI) for all references where available.
  • Lines 374-436: Please also check the formatting of the references (e.g., spaces in front of the list of authors: refs 17,18,23,28; adding a new line within a given reference: ref 2).

General:

  • Lines 360-366: Can you provide the number or code of the approval of the local ethics committee?
  • Lines 369-370: Important: Data should be made available to the reviewers and to the readers, too. Please consider using an open access repository for data storage.
  • Please re-check spelling/ language throughout the manuscript and formatting (e.g., spaces in table 4).

Author Response

Dear Editor and Reviewers,

Thank you very much for your kind and valuable comments. All changes in the manuscript

were highlighted in text.

Kind regards

The authors

Reviewer 1

Thank you for the invitation to review for IJERPH. The submitted manuscript examines the effects of in-season contrast strength training in male youth volleyball players. The main outcomes comprise physical fitness variables (dynamic balance, strength, power) and lower limb asymmetry. The results demonstrate greater improvements for the intervention group, indicating its benefit when performed in addition to volleyball training. Overall, this is a well-written and interesting paper that might fit well within the scope of the Special Issue "Training for Optimal Sports Performance and Health". However, I have some major and several minor remarks regarding the manuscript and look forward to receiving a revised version by the authors.

Dear reviewer, we thank you for your valuable comments and all your comments were considered word by word in the article and all of them are answered below.

General/ major comments:

  1. I recommend that data should be made available to the reviewers and to the readers, too. Please consider using an open access repository for data storage.

Response: The authors would like to thanks the reviewer for the valuable comment/suggestion. Therefore, we are using an open access repository for data storage and the row data are now free for readers and reviewers. Thank you.

  1. It seems that no compulsory adjustments for multiple comparisons have been made in the statistical analyses (e.g., Bonferroni corrections to avoid alpha error accumulation). This must be addressed when revising the manuscript.

Response: The authors would like to thanks the reviewer for the valuable comment and suggestion. As you recommend, we have reworked the statistical analysis to include bonfoeeroni adjustments ot p values and have also now included partial eta2 as a measure of effect size in analysis of the data and the result section are now revised and rewritten. Thank you.

  1. It would be helpful if the authors could add a documentation of the procedures (particularly the Y-Balance test) as supplementary material.

Response: Authors would like to thanks the reviewer for his/her comment. The Maximal reaching distances were recorded to the nearest 0.5-cm marker on the Y-balance test kit. Thank you. This statement was added in the procedure section as follow:

“The Maximal reaching distances were recorded to the nearest 0.5-cm marker on the Y-balance test kit”.

  1. Using the reference provided below, I recommend a check whether the appropriate formula was used for the calculation of lower limb asymmetry.

Response: Authors would like to thanks the reviewer for his/her comment. As you recommend, and according to the Parkinson et al. (2021) study, it seems that index nr. 10 is more suitable for calculating the lower limb asymmetry between the two legs during the single leg hop test in athletes. Accordingly, the formula of Parkinson et al. (2021) was used with the present results as follow: (Right leg - Left leg) / (Right leg + Left leg) × 100. Thank you. Changes are highlighted in the procedure section as follow:

“In the literature [19], relative inter-limb asymmetry for the lower limbs was determined according to the Parkinson et al. (2021) study and using the formula: (Right leg - Left leg) / (Right leg + Left leg) × 100. With this formula, it seems that index nr. 10 is more suitable for calculating the lower limb asymmetry between the two legs during the single leg hop test.”

Abstract:

  • Line 27 (and throughout the manuscript):Regarding the variables examined, I suggest being more specific: The YBT is a dynamic balance test - not just a (static) single-leg stance. Therefore, my recommendation is to rephrase “balance” to “dynamic balance”, also consistently throughout the manuscript.

Response: Authors would thank the reviewer for his/her valuable comments. Accordingly, the composite score during the Y-balance test was used to define the dynamic balance performance. In this context, we are using dynamic balance through the manuscript instead of balance only. Changes are highlighted in the text. Thank you.

  • Line 26-29 (and throughout the manuscript): Likewise, the authors categorize their variables by referring to the various motor abilities (dynamic balance, power…). Here, I recommend rephrasing the following sentence to become more specific:

“The tests were performed before and after 8-week training including dynamic balance (composite score during-balance test [CS-YBT]), maximum strength (one repetition maximum back squat, 1RM), explosive power (single leg hop [SLH] or countermovement jump height [CMJ]), as well as lower-limb asymmetry predicted from the single-leg jump performance between two legs [ILA]).”

Response: Authors would like to thank the reviewer for his/her valuable comment. As you recommend we are rephrasing the mentioned sentences to became more specific. The statement was revised as follow:

“Testing was conducted before and after the 8-week training programs and included the CS-YBT, 1 RM, SLHR and SLHL, CMJ and ILA.” 

  • Lines 30-31: Here, p-values with four decimal places are given; however, three decimals are sufficient. In the results section, only two decimals are presented. Please be consistent with three decimals for p-values throughout the manuscript.

 Response: Authors would like to thank the reviewer for his/her valuable comment. As you recommend we are using p-values with three decimals through the manuscript. Thank you.

Introduction

  • Line 66:Please elaborate on the term neuromuscular inter-limb asymmetries. What is meant here specifically, and what exactly was examined in the cited studies? In the current version, this does not become entirely clear to the readers. This is important, as these explanations lead to the objectives of the study.

Response: Authors would like to thank the reviewer for his/her valuable comment. According to the relevant literature (Parkinson et al., 2021, Bishop et al., 2019, Dos' Santos et al., 2018) We are referring to the wording “lower or inter-limb asymmetry” in general to define the inter-limb asymmetry. The statement was inserted in the introduction section as follow:

“In addition to performance, lower limb asymmetries have frequently been studied to quantify performance differences between limbs [9-11-19] and a bilateral asymmetry in youth.”

Materials and Methods

  • Lines 96-97: Please explain, which kind of randomization has been carried out.

Response: Authors would like to thank the reviewer for his/her valuable comment. In the present study, participants were randomly allocated to three experimental groups and a control. Group allocation was realized by adjusting for, age, maturation and their performance in the CMJ and 1 RM of the study sample. In addition, the order of each trial was changed randomly between participants, in order to avoid learning effects and fatigue. This kind of randomization was more explained in the procedure section. Thank you.

  • Line 171:In this respect, the authors indicate that groups were matched for anthropometric and physical characteristics. Please specify this procedure and describe it in more detail – thank you.

Response: Authors would like to thank the reviewer for his/her valuable comment. As we mentioned above, participants were randomly allocated to three experimental groups and a control. Group allocation was realized by adjusting for, age, maturation and their performance in the CMJ and 1 RM of the study sample. In addition, the order of each trial was changed randomly between participants, in order to avoid learning effects and fatigue. The later was more explained and highlighted in the study design. Thank you.

  • Line 100:How did you assess “socioeconomic backgrounds” – and is this likely to affect the results of the study? In case it is important to mention, please indicate a) how it was assessed, and b) the results of the assessment.

Response: Authors would like to thank the reviewer for his/her valuable comment. In fact, we are referring to describe the sample studied participant with regards to their samely school and volleyball practice schedules or the training experience. This is for us more important than their scio-ecomnomic background. Thank you. This is more explained in the participant subsection as follow:

“All participants had the same daily school and volleyball team-training schedules. They all had been playing volleyball on a regular basis three-four times a week (i.e., 90 min per session) with a match played during the weekend for more than 3 years.”

  • Lines 95-111:Please move all results (means & SD, table 1, ICC) to the results section.

Response: Authors would like to thank the reviewer for his/her valuable comment. As you recommend, all results (means & SD, table 1, ICC) were removed from the results section. Thank you.

  • Line 122: Is [13] the correct reference, here? Please check.

Response: Authors would like to thank the reviewer for his/her valuable comment. The corresponding reference was corrected to Fusco et al., (2020) [16] in the procedure section when describing the Y-BT protocol. Thank you.

  • Lines 125-126:When a measuring tape is used, I doubt that the Y-balance test was used, as this is performed on the Y-balance board, on which participants shove a box in three directions and the read the distance reached on the scale of the Y-balance board. Hence, no measuring tape is needed. Maybe the Star Excursion Balance Test was used? Please clarify, ideally using a figure to describe testing procedures (maybe as supplementary material) – thank you!

Response: Authors would like to thank the reviewer for his/her valuable comment. As you recommend a figure 1 was added in to the procedure section to describe the Y-Balance protocol and execution. Thank you.

  • Line 129: Please indicate how you measured the length of the leg?

Response: Authors would like to thank the reviewer for his/her valuable comment. According to Garisson et al. (2013), the leg length was determined using the distance between the most prominent portion of the greater trochanter and the floor while the individual was in a standing position. Thank you.

The later explanation was added in the procedure section as follow:

“The leg length was determined using the distance between the most prominent portion of the greater trochanter and the floor while the individual was in a standing position (Garrisson et al., 2013).”

  • Line 149:Please check the term “maximal height jump”. I think “jump for maximal distance” is more appropriate here.

Response: Authors would like to thank the reviewer for his/her valuable comment. As you recommend “maximal height jump” was replaced by “jump for maximal distance”. Thank you.

  • Lines 155-161:For the assessment of bilateral asymmetry, the formula
    (A-B) / A x 100 is used and an appropriate reference is provided. However, I’d recommend consulting the paper by Parkinson et al. 2021: https://www.jssm.org/jssm-20-594.xml%3Eabst, particularly Table S1; to verify whether the formula used by the authors, corresponding to index nr. 7 in the above-mentioned paper, is more appropriate, or whether index nr. 10 seems more suitable for their study.

Response: Authors would like to thanks the reviewer for his/her comment. As you recommend, and according to the Parkinson et al. (2021) study, it seems that index nr. 10 is more suitable for calculating the lower limb asymmetry between the two legs during the single leg hop test. Accordingly, the formula of Parkinson et al. (2021) was used with the present results as follow: (Right leg - Left leg) / (Right leg + Left leg) × 100. Thank you. Changes are highlighted in the procedure section as follow:

“In the literature [19], relative inter-limb asymmetry for the lower limbs was determined according to the Parkinson et al. (2021) study and using the formula: (Right leg - Left leg) / (Right leg + Left leg) × 100. With this formula, it seems that index nr. 10 is more suitable for calculating the lower limb asymmetry between the two legs during the single leg hop test.”

  • Lines 204-205:Please add the categories for ICC interpretation (e.g. low, moderate, high, excellent).

Response: Authors would like to thanks the reviewer for his/her comment. According to Portney, (2000), an ICC greater than 0.80 reflects excellent reliability, whereas ICCs from 0.70 to 0.79 reflect good reliability. Thank you.

The following statement was added to the statistical procedure section as follow:

“For the interpretation of ICC values, a value greater than 0.80 reflects an excellent reliability, whereas ICCs from 0.70 to 0.79 reflect a good reliability (Portney et al., 2000).”

  • Line 168:In this respect, an ICC value of 0.89 indicates good reliability, and coefficients above 0.90 indicate excellent reliability? Please re-check.

Response: Authors would like to thanks the reviewer for his/her comment. According to the classification of Portney et al. (2000), we have checked and corrected the interpretation of the ICC values through the procedure section. Thank you.

  • Lines 202-203:What is the rationale for converting effect sizes from partial eta-squared to Cohen’s d? Partial eta-squared is a universally recognized/ accepted effect size, too. Please clarify.

Response: The reviewer makes a good point. We have changed the effect size and now present partial eta2.

  • Lines 195-207:It seems that no corrections for multiple comparisons (e.g. Bonferroni adjustments) have been applied. However, this seems to be necessary due to multiple testing. Please re-check and either apply these adjustments in the statistical analyses or – if already done – please indicate which procedure was used to avoid alpha error accumulation.

Response: Yes, good point. We now make it explicit that we undertook post-hoc pairwise comparisons with Bonferroni adjustment

Results

  • Lines 248 & 258: Is it possible to improve the quality of the figures? Furthermore, effect sizes are presented as dand also as ES; and decimals of p-values vary from 2 to 3 places; but should be consistent.

Response: Authors would like to thanks the reviewer for his/her comment. The new figure 2 and 3 are now included and consistent with a effect size and p-value; and re-inserted therefore in the text.

Discussion

  • Lines 279-280:Please re-check “With children’s, balance and coordination are less developed [23].

Response: Authors would like to thanks the reviewer for his/her comment. As you recommend the mentioned statement are reworded as follow:

“While, balance and coordination are less developed in children’s [23], greater enhancement of dynamic balance performance with contrast strength exercises could lead to a greater muscle power output”. Thank you.

  • Lines 284-286:I recommend rephrasing this sentence more cautiously. This might be an explanation for the findings of this study. However, “stress on the muscle spindles” was not subject to the investigation. At least, a reference supporting this statement/ explanation should be added.

Response: Authors would like to thanks the reviewer for his/her comment. We agree with you that stress on the muscle spindles” was not subject to the investigation. Therefore, we were rephrasing this explanation as follow:

“In the current study, CST was performed under less stable conditions with high-speed dynamic contractions performed within a more limited base of support or with the center of gravity being moved outside the base of support, which would be affected to a much greater extent by balance and strength/power output”. Thank you.

  • Line 307:Again, I recommend the paper by Parkinson et al. 2021., who caution against the use of arbitrary asymmetry thresholds, such as 10-15%. I suggest adding a subordinate clause, referring to the failure to reference the origin of the evidence for an asymmetry threshold. Rather, more individual approaches to asymmetry have been proposed in recent studies recognizing the task-, metric- and population-specific nature of asymmetry.

Response: Authors would like to thanks the reviewer for his/her comment. We agree with you to adopt an adequate method to detect the lower limb asymmetry rather than magnitude of threshold such as referring to the failure to reference the origin of the evidence for an asymmetry threshold. Therefore, a new paragraph was added in the discussion section to more explain this as follow:

“However, failure to reference the origin of the evidence for an asymmetry threshold reinforces doubt over the use of arbitrary thresholds, such as 10-15% (Parkinson et al., 2021). Therefore, an individual approach to defining asymmetry may be necessary to refine robust calculation methods and to establish appropriate thresholds across youth athletes and methodologies that enable appropriate conclusions to be drawn. This is more important than an individual approach to asymmetry that have been proposed recognizing the task-, metric- and population-specific nature of asymmetry (Parkinson et al., 2021).”

References:

  • Lines 374-436:Please include the digital object identifier (DOI) for all references where available.

Response: Authors would like to thanks the reviewer for his/her comment. As you recommend, the DOI for all references where now available and highlighted in the references list. Thank you.

  • Lines 374-436:Please also check the formatting of the references (e.g., spaces in front of the list of authors: refs 17,18,23,28; adding a new line within a given reference: ref 2).

 Response: Authors would like to thanks the reviewer for his/her comment. As you recommend, all of the mentioned references were corrected. Thank you.

General:

  • Lines 360-366: Can you provide the number or code of the approval of the local ethics committee?

 Response: Authors would like to thanks the reviewer for his/her comment. Before experimental testing, the study was conducted according to the Declaration of Helsinki, and the protocol was fully approved by the Ethics Committee of the National Centre of Medicine and Science of Sports of Tunis (CNMSS-LR09SEP01) before the commencement of the assessments. The statement was now added in the procedure section. Thank you.

  • Lines 369-370:Important: Data should be made available to the reviewers and to the readers, too. Please consider using an open access repository for data storage.

Response: The authors would like to thanks the reviewer for the valuable comment/suggestion. Therefore, we are using an open access repository for data storage and the row data are now free for readers and reviewers. Thank you.

  • Please re-check spelling/ language throughout the manuscript and formatting (e.g., spaces in table 4).

Response: The authors would like to thanks the reviewer for the valuable comment/suggestion. The spelling/ language in the whole manuscript was revised by a native speaker Pr. Michael Duncan and Prof Tom Bowmen. In addition, the table 2 was no cleaned. Thank you.

Reviewer 2 Report

General Comments:  Nice effort by the research team. The article addresses a great topic and a clear need for research within this area. The population studied was my biggest question. The information you provided within the literature is relevant and laid out nicely leading into your purpose statement. However, there are a few edits which need to be addressed to ensure the article is sound and ready for publication. My main concern are the conclusions made. You may need to soften the conclusions based on the number of subjects and the age of the subjects. The research team makes some pretty broad conclusions that probably need to be softened.

Title: change the title

Abstract: This is a very important part of your paper. The abstract gives a nice overlay of what is to come in the research article. However, much of the abstract is culminated of the results of the study. Think about having less result content and add some previous literature leading into your purpose statement. The abstract clearly states the purpose, results, and take-aways from your research, which is great.

Introduction: The introduction starts with a great lead in with background information to what the study involves, although it seems to jump right into volleyball. Despite giving a brief definition of the sport, the reader is left to question what it exactly is.

Methods: Ultimately, the methods remain largely the most questionable section in this study. The number of subjects is low and the age is a concern. Some revisions are needed.

During the experiment; what did the control group achieve??? How did you control the training load?

Do the two groups have experience with strength training with additional load?

Why we used post hoc ??? I think that you don’t needed

L161: ICC= 0.73, Table 3 ??? ICC > 0.8

Add more information in the training program section

Results:  The results flowed nicely and were easily readable.  Overall, the results section was broken up well, easy to follow, and was not choppy. Great job here.

Discussion: Nice job overall with the discussion section. The use of research to back up the findings from the current study was used appropriately throughout this section. Overall, the discussion section is written very well. My main concern is the age of the subjects and the conclusions made.

Limitations & Future Research: Most all research has some time of limitation. The limitations and future research section is well written. As many of the limitations listed are surface level, think deeper into anything you and your research team specifically did or did not do that may constitute as a limitation to the study. The number of subjects is low and the age is a concern. Some revisions are needed.

Conclusion: Be sure to include a conclusion to your study. This is where you bridge the gap to the practitioner. This section has not been adequately developed. It needs come beefing up!

References: The references look great.

Tables: The tables look great.

Author Response

Dear Editor and Reviewers,

Thank you very much for your kind and valuable comments. All changes in the manuscript

were highlighted in text.

Kind regards

The authors

Reviewer 2

General Comments:  Nice effort by the research team. The article addresses a great topic and a clear need for research within this area. The population studied was my biggest question. The information you provided within the literature is relevant and laid out nicely leading into your purpose statement. However, there are a few edits which need to be addressed to ensure the article is sound and ready for publication. My main concern are the conclusions made. You may need to soften the conclusions based on the number of subjects and the age of the subjects. The research team makes some pretty broad conclusions that probably need to be softened.

Dear reviewer, we thank you for your valuable comments and all your comments were considered word by word in the article and all of them are answered below.

Title: change the title

Response: The authors would like to thanks the reviewer for the valuable suggestion. As you recommend, the title have been changed to: “Effects of 8-Weeks In-Season Contrast Strength Training Program On Measures of Athletic Performance and Lower Limb Asymmetry in Male Youth Volleyball Players”. Thank you.

Abstract: This is a very important part of your paper. The abstract gives a nice overlay of what is to come in the research article. However, much of the abstract is culminated of the results of the study. Think about having less result content and add some previous literature leading into your purpose statement. The abstract clearly states the purpose, results, and take-aways from your research, which is great.

Response: The authors would like to thanks the reviewer for the valuable comment/suggestion. Therefore, a newly paragraph in the introduction of the abstract section was included as follow:

“Practicing/training strength tasks using high compared vs. low strengthening exercise such as contrast training (CST) seems to be beneficial as it addresses larger adaptive reserves in youth athletes”. Thank you.

Introduction: The introduction starts with a great lead in with background information to what the study involves, although it seems to jump right into volleyball. Despite giving a brief definition of the sport, the reader is left to question what it exactly is.

Response: The authors would like to thanks the reviewer for the valuable affirmative comment. Thank you.

Methods: Ultimately, the methods remain largely the most questionable section in this study. The number of subjects is low and the age is a concern. Some revisions are needed.

Response: The authors would like to thanks the reviewer for the valuable comment/suggestion. The participant involved in the present study were a high level youth athlete’s member of a national volleyball team in professional league 1. In addition, the sample was randomly allocating in to similar group matched from age, maturation and physical fitness. Hence a low number of 15 participant seems to sufficient to demonstrate a meaningful result with the present study. Thank you.

During the experiment; what did the control group achieve??? How did you control the training load?

Response: The authors would like to thanks the reviewer for the valuable comment/suggestion. As mentioned in the procedure section, the control group (CG) followed their standard volleyball practice over the same duration with no strength training design. Furthermore, we control the training load with regards to training principle that preconize that the load should be increased through the period of training with a tapering phase and that volume and intensity should be inversely proportional (Pradet et al., 1997).

A newly paragraph was included in the training program as follow:

“Participants of the control group (CG) followed their standard volleyball practice over the same duration with no strength training design”. Thank you.

Do the two groups have experience with strength training with additional load?

Response: The authors would like to thanks the reviewer for the valuable comment/suggestion. The two groups had regularly performed additional strength/power exercises during competitions and training for a minimum of 2 years before the start of the study. Therefore, a newly paragraph was inserted in the procedure section as follow:

“It is important to note that all participants from the two groups had regularly performed strength/power training exercises (i.e., bench press, pull over, squat, forward lunge snatch, and plyometric) during competitions and training for a minimum of 2 years before the start of the study.” Thank you.

Why we used post hoc ??? I think that you don’t needed

Response: The authors would like to thanks the reviewer for the valuable comment/suggestion. Recognizing there were no significant differences between any of the dependent variables at baseline between intervention (INT) and control (CON) groups, any changes in the dependent variables pre-post intervention were examined using a series of 2 (pre-post) X 2 (INT 19 vs CON) repeated measures analysis of ANOOVA). In addition, Partial Æž2 as a measure of effect size. Where any significant differences were found post hoc pairwise comparisons (Bonferroni adjusted) should be employed to examine where the differences lay. Thank you.

L161: ICC= 0.73, Table 3 ??? ICC > 0.8

Response: The authors would like to thanks the reviewer for the valuable comment/suggestion. As you recommend we revised the ICC value through the manuscript and as suggested by the reviewer 1 the table 3 of reliability analyses was moved. Thank you.

Add more information in the training program section

Response: The authors would like to thanks the reviewer for the valuable comment/suggestion. As you recommend a more information with regards to the above concerns in the training program was added. Thank you.

Results:  The results flowed nicely and were easily readable.  Overall, the results section was broken up well, easy to follow, and was not choppy. Great job here.

Response: The authors would like to thanks the reviewer for the valuable affirmative comment. As suggested by the reviewer 1 the result section was rewritten. Thank you.

Discussion: Nice job overall with the discussion section. The use of research to back up the findings from the current study was used appropriately throughout this section. Overall, the discussion section is written very well. My main concern is the age of the subjects and the conclusions made.

Response: The authors would like to thanks the reviewer for the valuable affirmative comment. As you recommend, we include a short paragraph in the study limitation to reporting the age and the small sample recruited in the study. In addition, the conclusion was improved as follow:

“This suggests that there could be interdependent positive transfer effects, from training that is singularly focused on strength and conditioning. Despite this, the present findings are based on longitudinal data which does conclusively allow for cause-and-effect relationships to be determined. Furthermore, the findings suggest that when designing training programmes aimed to improve athletic performance in volleyball players, coaches should pay attention to these specific adaptation, which can necessitate an individualized approach to programme design”. Thank you.

Limitations & Future Research: Most all research has some time of limitation. The limitations and future research section is well written. As many of the limitations listed are surface level, think deeper into anything you and your research team specifically did or did not do that may constitute as a limitation to the study. The number of subjects is low and the age is a concern. Some revisions are needed.

Response: The authors would like to thanks the reviewer for the valuable comment. Accordingly, a short paragraph in the limitation of the study was inserted as follow:

Finally, the sample size of each groups was small. Therefore, this study is preliminary. However, it is difficult and almost impossible to recruit large sample sizes in elite sport, especially in a highly professionalized elite sport such as volleyball. While our results provide interesting information for coaches and strength and conditioning specialists, they have to be interpreted with caution and should be verified in future studies.

Conclusion: Be sure to include a conclusion to your study. This is where you bridge the gap to the practitioner. This section has not been adequately developed. It needs come beefing up!

Response: The authors would like to thanks the reviewer for the valuable comment. As you recommend, a newly paragraph was added in the conclusion section to more clarify the importance of the present study for the reader; as follow:

“This suggests that there could be interdependent positive transfer effects, from training that is singularly focused on strength and conditioning. Despite this, the present findings are based on longitudinal data which does conclusively allow for cause-and-effect relationships to be determined. Furthermore, the findings suggest that when designing training programmes aimed to improve athletic performance in volleyball players, coaches should pay attention to these specific adaptation, which can necessitate an individualized approach to programme design”. Thank you.

References: The references look great.

Response: The authors would like to thanks the reviewer for the valuable comment. Thank you.

Tables: The tables look great.

Response: The authors would like to thanks the reviewer for the valuable comment. As suggested by the reviewer 1 table are revised through the manuscript. Thank you.

Round 2

Author Response

Dear Editor and reviewer

We thank all the reviewers for accepting the article. The comments of the remaining respected reviewer was also carefully followed below and corrected in the article.

Best regards

Hadi Nobari Phd

Reviewer 1

Thank you for the extensive changes that have been made to improve clarity and understanding, but unfortunately only in parts of the manuscript. There are still critical points to note, and some arose additionally during the revision process, as outlined below:

Dear reviewer, thank you very much.

MAJOR #1

Data availability:

The authors responded to the first comment that they are using an open access repository for data Data Availability Statement ays: The data presented in this study are available on reasonable request 477 from Abdeltif Mesfar. -478). Please indicate where the data can be accessed!

Response: The authors would like to thanks the reviewer for the valuable comment/suggestion. We now indicate in the data availability statement that any individual interested in access of the data should email the first author.

#2 Regarding the Y-Balance Test:

The author states that the leg length was determined using the distance between the most prominent portion of the greater trochanter and the floor while the individual was in a standing position according to Garisson et al. (2013).

Unfortunately, I cannot find the appropriate reference. In their review response, the authors refer to Garrisson et al. 2013, however, this reference deals with ulnar collateral ligament tears and is in no way related to the present study. In the references, Garrisson et al. 2013 appear as ref [15], but this reference does not appear in the manuscript! Additionally, regarding the determination of leg length, the authors refer to ref [17] (line 139), which seems incorrect: Ramírez-Campillo et al. 2015: Effect of vertical, horizontal, 519 and combined plyometric training on explosive, balance, and endurance performance of young soccer players. => Please recheck and clarify!

Response: The authors would like to thanks the reviewer for the valuable comment/suggestion. We agree with reviewer that the cited reference (Garrisson et al. 2013) and the description method of leg length were not appropriate with the present study. Therefore, the reference Garisson et al., 2013 was moved from the text and the reference section and the appropriate method to measure the lower limb length was described by Fusco et al. 2020 as follow:  Lower limb length was measured from the anterior superior iliac spine to the most distal part of the medial malleolus by using a tape measure while the subject laid in supine position. In addition, we are revising and recheck all the reference in the text and they are now in line with the order of the reference section. Thank you.

In the same paragraph, Fusco et al. (2020) are cited, and this is an appropriate reference. Moreover, in their study they describe the correct procedure for measuring leg length for use with the Y-Balance-test. At the bottom of page 2 in their paper, left column, it says: Lower limb length was measured from the anterior superior iliac spine to the most distal part of the medial malleolus by using a tape measure while the subject laid in supine position. This differs from the procedure used by the authors in the present study. Therefore, if possible, the correct measurements should be made up afterwards. If this is no longer possible, this deviation from the standard procedures must be pointed out in the manuscript!

Response: The authors would like to thanks the reviewer for the valuable comment/suggestion. We agree with the you with the appropriate method to determine the leg length as described by Fusco et al. 2020. Accordingly, Lower limb length was measured from the anterior superior iliac spine to the most distal part of the medial malleolus by using a tape measure while the subject laid in supine position. This description was inserted in the text. Thank you.

Please also add information regarding the stance foot: were participants allowed to lift the heel during the anterior reaching maneuvers? This is not recognizable in figure 1, and also varies across different studies published.

Response: The authors would like to thanks the reviewer for the valuable comment/suggestion. During the Y-Balance test protocol, all trials were conducted barefooted and participants did not allow to lift the heel during the anterior reaching maneuvers according to Kang et al., 2015. Because, the allowance of heel lift could reduce the importance of ankle dorsiflexion, while requiring the heel to maintain contact with the ground would emphasize ankle range of motion (Nelson et al., 2021).  Therefore, the description of the Y-Balance test protocol was now rewritten in the text. Thank you.

Spelling/ language/ formatting

The authors indicated that spelling/ language in the whole manuscript was revised. Nevertheless, even while reading the abstract, I noticed several issues that should be clarified, e.g.

Line 25: I guess the authors refer to high and low loads (as outlined in their introduction and methods), instead of upper vs. lower strength training (=> do you mean upper and lower body?).

Response: The authors would like to thanks the reviewer for the valuable comment/suggestion. We mean high and load strength exercise during training program. The statement was corrected as follow: “Strength training using high and lower load such as contrast training (CST) seems to be beneficial as it addresses larger adaptive reserves in youth athletes.” Thank you.

Additionally, either use compared to “vs”. Furthermore, be straightforward: better start with strength training using…

Response: The authors would like to thanks the reviewer for the valuable comment/suggestion. The statement was corrected as follow: “Strength training using high and lower load such as contrast training (CST) seems to be beneficial as it addresses larger adaptive reserves in youth athletes. Thank you.

Lines 31-33: The intervention group (N = 16; age, 14.4±0.6 31 years) performed CST twice weekly (half-squat, lunge, bench press, and pullover) with the same set, rest, and movements.

Response: The authors would like to thanks the reviewer for the valuable comment/suggestion. The statement was corrected as follow: “Thirty-one male youth volleyball players aged 14 years were randomly assigned to a CST group (n=16) or a control group (n=15)”. Thank you.

Line 34: The word including should be removed

Response: The authors would like to thanks the reviewer for the valuable comment/suggestion. Removed. Thank you.

Line 34-35: Results revealed a significant group × time interactions were observed Singular or plural Results revealed significant group × time interactions significant group × time interactions were observed”

Response: The authors would like to thanks the reviewer for the valuable comment/suggestion. Therefore, spelling was corrected in the text as follow:  Significant group × time interactions was observed for CS-YBT [p< 0.001, ηp2 = 0.70], 1RM [p< 0.001, ηp2 = 0.95], SLH with right and left leg [p< 0.001, ηp2 = 0.69 and 0.51], CMJ [p< 0.001, ηp2 = 0.47]), whilst it was not notable in ILA [p< 0.294]. Thank you.

Line 39: Is “timing” the correct term, or you mean training? Please recheck

Response: The authors would like to thanks the reviewer for the valuable comment/suggestion. Yes, training not timing. Corrected. Thank you.

Regarding formatting,

 p-values still vary in their decimals (page 6-page9; text, figure 1, figure 2, table 2). Likewise, table 2: 1 or 2 decimals for means and SD are sufficient!

Response: The authors would like to thanks the reviewer for the valuable comment/suggestion. We originally included actual P values. However, we have now adjusted to use only 2 decimal places as suggested

As to the remaining text, I honestly doubt that the manuscript was proofread by two native speakers. Too many phrases remain unclear to the reader, e.g.

Lines 379-380: The main findings were that the CST group improved in 379 all physical fitness parameters but not the lower limb asymmetry, while the CG did not. Please specify

!

Response: The authors would like to thanks the reviewer for the valuable comment/suggestion. We have modified the sentence and hope it now reads with better clarity

Lines 383-384: While balance and coordination are less developed in children’s [24] …please rephrase

And many more…

Response: The authors would like to thanks the reviewer for the valuable comment/suggestion. We have rephrased the statement and hope it now reads more effectively.

Contrary, the conclusion has been completely revised and reads very, very well!!! Please continue this good work and apply it to the whole manuscript.

Response: The authors would like to thanks the reviewer for the valuable comment/suggestion. 

MINOR:

 Lines 96-97: participants were randomly allocated to three experimental groups and a control. Three experimental groups? Please recheck!

Response: The authors would like to thanks the reviewer for the valuable comment/suggestion. Corrected. Thank you.

Line 139-140: An excellent reliability score was reported for the CS-Y balance test with the 139 intra class correlation coefficient (ICC) value of 0.93 provide a reference or otherwise clarify that this ICC originates from your results (as you did for 1RM in lines 169-170).

Response: The authors would like to thanks the reviewer for the valuable comment/suggestion. The ICC value was reported in the present study. Corrected thank you.

Line 237: Again, references are incorrect. It seems that [19] should be [20]? Carefully check references throughout the manuscript!

Response: The authors would like to thanks the reviewer for the valuable comment/suggestion. All references are rechecked in the whole manuscript. Thank you.

Lines 379-380: Not correct, according to the text in the results section. The CG improved also improved in most variables, but to a lesser degree. => Again, it would be very helpful if the authors could provide their study data!

Response: The authors would like to thanks the reviewer for the valuable comment/suggestion. We uploaded a data as a supplementary file.

Another co-author has joined but Author Contributions -466). Please clarify!

Response: The authors would like to thanks the reviewer for the valuable comment/suggestion. The contribution of Professor Michael Duncan was included. Thank you.

Line 74: Reference is missing please clarify

Response: The authors would like to thanks the reviewer for the valuable comment/suggestion. The reference was included. Thank you. Thank you.

Lines 107-108: please add the number of athletes in each group

Response: The authors would like to thanks the reviewer for the valuable comment/suggestion. The number of athletes was included. Thank you.
